# Three-Dimensional Printing of a Container Tablet: A New Paradigm for Multi-Drug-Containing Bioactive Self-Nanoemulsifying Drug-Delivery Systems (Bio-SNEDDSs)

**DOI:** 10.3390/pharmaceutics14051082

**Published:** 2022-05-18

**Authors:** Vineet R. Kulkarni, Mohsin Kazi, Ahmad Abdul-Wahhab Shahba, Aakib Radhanpuri, Mohammed Maniruzzaman

**Affiliations:** 1Pharmaceutical Engineering and 3D Printing (PharmE3D) Labs, Division of Molecular Pharmaceutics and Drug Delivery, College of Pharmacy, The University of Texas at Austin, Austin, TX 78705, USA; vineetkulkarni@utexas.edu (V.R.K.); a.radhanpuri@austin.utexas.edu (A.R.); 2Department of Pharmaceutics, College of Pharmacy, King Saud University, P.O. Box 2457, Riyadh 11451, Saudi Arabia; mkazi@ksu.edu.sa (M.K.); shahba@ksu.edu.sa (A.A.-W.S.)

**Keywords:** 3D printing, poor solubility, SNEDDS, multi-drug delivery, fused deposition modeling, bioactive

## Abstract

This research demonstrates the use of fused deposition modeling (FDM) 3D printing to control the delivery of multiple drugs containing bioactive self-nano emulsifying drug-delivery systems (SNEDDSs). Around two-thirds of the new chemical entities being introduced in the market are associated with some inherent issues, such as poor solubility and high lipophilicity. SNEDDSs provide for an innovative and easy way to develop a delivery platform for such drugs. Combining this platform with FDM 3D printing would further aid in developing new strategies for delivering poorly soluble drugs and personalized drug-delivery systems with added therapeutic benefits. This study evaluates the performance of a 3D-printed container system containing curcumin (CUR)- and lansoprazole (LNS)-loaded SNEDDS. The SNEDDS showed 50% antioxidant activity (IC_50_) at concentrations of around 330.1 µg/mL and 393.3 µg/mL in the DPPH and ABTS radical scavenging assay, respectively. These SNEDDSs were loaded with no degradation and leakage from the 3D-printed container. We were able to delay the release of the SNEDDS from the hollow prints while controlling the print wall thickness to achieve lag phases of 30 min and 60 min before the release from the 0.4 mm and 1 mm wall thicknesses, respectively. Combining these two innovative drug-delivery strategies demonstrates a novel option for tackling the problems associated with multi-drug delivery and delivery of drugs susceptible to degradation in, i.e., gastric pH for targeting disease conditions throughout the gastrointestinal tract (GIT). It is also envisaged that such delivery systems reported herein can be an ideal solution to deliver many challenging molecules, such as biologics, orally or near the target site in the future, thus opening a new paradigm for multi-drug-delivery systems.

## 1. Introduction

The past few years have seen an increasing interest in using three-dimensional printing for pharmaceutical research and the development of personalized and patient-specific drug-delivery systems [1]. Such personalized systems can deliver better and more efficient therapeutic doses, while minimizing the possible side effects and adverse reactions [2]. Additionally, innovative research on 3D printing has further gained an impetus after the approval of Aprecia Pharmaceutical’s SPRITAM^®^ from the US FDA in 2015 [3]. This was followed by the recent Investigational New Drug (IND) 505(b)(2) application clearance for China-based Triastek, Inc.’s (Nanjing, China), a 3D-printed drug product—T20; the use of 3D printing in pharmaceuticals will be explored further with more vigor [4,5]. Various applications of different 3D-printing platforms, such as fused deposition modeling (FDM), selective laser sintering (SLS), stereolithographic 3D printing (SLA), binder-jetting 3D printing, and ink-jet 3D printing, have been explored [6]. These technologies have been used at an individualistic level to develop pills and films and other dosage forms for various drugs across the therapeutic classes [7]. The incorporation or use of these processes in conjunction with other drug-delivery platforms would further help to open a broad range of avenues and arenas to develop and formulate dosage forms in a multitude of different ways.

The development of lipid-based carriers has gained a lot of traction with a few products already being available on the US market [8]. These help to deliver poorly water-soluble drugs in their dissolved state, and the relatively smaller droplet size of the lipid system, which provides a large surface area and promotes a faster release and better drug absorption [8,9]. Self-emulsifying drug-delivery systems (SEDDS), self-micro-emulsifying drug-delivery systems (SMEDDS), and self-nano-emulsifying drug-delivery systems (SNEDDS) are some of the most widely used lipid-based drug-delivery systems [10,11]. These systems are isotropic mixtures of an active drug compound with lipids, surfactants, and/or water-soluble co-solvents that produce ultrafine emulsions upon gentle agitation in aqueous phase, such as the upper GI lumen content [1]. SEDDS, SMEDDS, and SNEDDS can be differentiated basically according to their droplet size upon aqueous dispersion [1]. SEDDS spontaneously forms emulsions (translucent) with a droplet size ranging from 250 nm to 1.0 μm [1]. SNEDDSs are isotropic mixtures of oil, surfactant (HLB > 12), and co-surfactant that spontaneously form nano-emulsions with a droplet size of 100–250 nm [2]. In comparison, SMEDDSs contain a higher content of hydrophilic surfactants and cosurfactants, wherein the lipid content is reduced. SMEDDSs form isotropic and thermodynamically stable microemulsions, with a droplet size of less than 100 nm [2]. Around two-thirds of the new chemical entities being introduced in the market are associated with some inherent issues, such as poor solubility and high lipophilicity [12]. The SEDDS/SMEDDS/SNEDDS provide a good and easy way to develop a platform for such drugs [8,9,10,11].

Drug precipitation and phase separation are some of the major issues associated with the conversion of formulated liquid SNEDDSs to easy-to-handle solid SNEDDSs. Other challenges, such as poor palatability due to high lipid compositions and poor transportability, also need to be addressed [12,13].

SNEDDSs are inherently associated with a burst release and face the challenge of dose dumping in case of a poorly developed system [14]. It is difficult to deliver SNEDDSs while controlling their release, thus making this highly effective technology a poor option for the sustained or controlled release for treating GIT issues [15].

Combining the two drug-delivery platforms, that is, 3D printing and SNEDDSs, would further aid in developing new strategies for developing poorly soluble drugs while overcoming the above-mentioned challenges, and open new avenues for personalized drug delivery with added therapeutic benefits.

The FDM 3D-printing platform can be used to design and manufacture hollow tablets of various geometries, which can take the shape of a container [16]. Additionally, these hollow container systems can be designed to have a drug release, based on the filament composition used to print them [17]. Using hydrophilic polymers, such as hydroxypropyl cellulose (HPC), along with super-disintegrants would aid in obtaining a faster release as opposed to using pH-dependent polymers, such as HPMCAS grades, which form a hydrogel-like structure upon swelling [18]. Additionally, the release profiles can be further optimized by altering the geometry of the designs, for example, modifying the wall thickness or in-fill density [19]. The selection of printing parameters, such as the printing temperature, print bed temperature, print speed, and material flow rate, determines the quality of the print and influences the dimensional accuracy of the final print. Optimizing these is essential to ensure that the layers adhere to each other well and do not collapse over each other [19]. Incorporating the pre-formulated SNEDDSs into these container systems could aid in delivering the former to the gastrointestinal tract (GIT) without the commonly associated issue of degradation and precipitation during the transit usually associated with the super-saturation of the GI fluids. It would also help to deliver the SNEDDS over a longer period, as opposed to having an immediate release effect. This would help to effectively treat diseased conditions, such as ulcerative colitis, inflammatory bowel disorder, and Crohn’s disease [20,21].

More studies need to be conducted to explore the incorporation of SNEDDSs into 3D-printed tablets [22,23,24]. In the current study, we report the incorporation of the pre-formulated SNEDDSs into 3D-printed-container systems, for the delivery of two different drugs with inherent poor solubility, namely, curcumin (CUR) and lansoprazole (LNS). Both are indicated for their action in treating duodenal and gastric ulcers. These SNEDDSs would be incorporated into a hollow tablet, printed using FDM 3D-printing technology. These container systems would then be evaluated to assess the effect of combining these two delivery platforms and the efficiency of design-dependent drug releases.

## 2. Materials and Methods

### 2.1. Materials

Lansoprazole (LNS, purity = 99%) and curcumin (CUR, purity = 99.5% pure) were purchased from Enzo Life Sciences, (Lausen, Switzerland). Black seed oil (BSO) was cold-pressed, and its methods of collection, extraction, and isolation were explained in detail in our previous publication [25]. Capmul MCM (CMCM) (medium-chain mono and diglycerides C8-10) was obtained from Abitec Corp., Janesville, WI, USA. Transcutol P (TCP) and Kolliphor ELP (KELP) were obtained from Gattefosse, Saintpreist, France, and BASF, Ludwigshafen, Germany, respectively.

Ashland Inc., Wilmington, DE, USA donated Klucel^TM^ Hydroxypropyl cellulose (HPC) JF grades. Soluplus^®^ was a gift from BASF, Ludwigshafen, Germany. All other chemicals and solvents used in the study were of analytical grade or high-performance liquid chromatography (HPLC) grades.

### 2.2. Preparation of SNEDDS Formulations

Blank anhydrous SNEDDSs were prepared by weighing the oil, surfactant, and cosolvent in predetermined ratios presented in Table 1. A total of 25% *w*/*w* of BSO (long-chain fatty acids, C16–20) with 20% *w*/*w* Capmul MCM (CMCM, mono and diglycerides oil) were mixed with 5% *w*/*w* co-solvent and 50% *w*/*w* surfactant. The formulation was vortexed to ensure it was well-mixed. The anhydrous SNEDDSs were homogeneous along with their transparent appearance. The drug-loaded SNEDDSs were prepared by dissolving the model drugs, such as CUR and LNS, in blank formulations [25,26]. The right amount of drugs was estimated from the subsequent equilibrium solubility studies.

### 2.3. Equilibrium Solubility Studies

Excess CUR and LNS powder were added into the representative blank SNEDDS formulation to determine the drug solubility. Briefly, SNEDDSs with excess drugs were vortexed and kept in the incubator for 3 days at 37 °C. These formulations were centrifuged at 13,500 rpm (12,225× *g*) for 10 min and the supernatant was carefully separated and diluted for analysis with an appropriate solvent [25].

### 2.4. SNEDDS Self-Emulsification Tests, Droplet Size, and Potential Measurement

The self-emulsifying properties of each formulation were visually assessed after dilution in Milli-Q water at a (1:1000 *w*/*w*) ratio and evaluated for their appearance and homogeneity, which was further evaluated using droplet-size measurements [25]. The droplet size and polydispersity index (PDI) of the SNEDDSs were determined using a dynamic light-scattering (DLS) particle-size analyzer (Malvern Zetasizer Nano ZS, Model ZEN3600, Worcester-200 shire, UK) with a 633 nm laser. A total of 100 mg of anhydrous SNEDDSs was diluted with 100 mL Milli-Q water (1:1000 *w*/*w* dilution) before droplet-size measurements were obtained. The droplet size of the aqueous dispersion (nano-emulsion) was determined by DLS at a scattering angle of 173° and 25 °C. The Zeta potential of the SNEDDSs was evaluated by the laser doppler velocimetry (LDV) mode using the same Nano ZS at 25 °C. The average particle size, polydispersity index, and zeta potential were repeated three times, and the average values were used. Ten measurements were obtained as set by the method of the software [25].

### 2.5. Transmission Electron Microscopy (TEM)

The microscopic characterization of the SNEDDSs was performed using JEOL JEM1010 transmission electron microscopy (JEOL Ltd., Akishima, Tokyo, Japan). The sample was freshly prepared by putting it on the carbon-coated copper grid and osmium was used to stain the lipid components. The dry sample was loaded into the TEM and viewed at different magnification levels [26].

### 2.6. Antioxidant Studies

#### 2.6.1. DPPH Radical Scavenging Activity Assay

The antioxidant activity of the drug-free SNEDDSs was estimated by the DPPH (2,2-diphenyl-1-picrylhydrazyl) radical scavenging assay, as reported by Fahad et al., 2020 with minor modifications [27,28,29]. Various concentrations (10, 50, 100, 500, and 1000 μg/mL) of the representative drug-free SNEDDSs were prepared. Thereafter, 0.5 mL of each concentration was mixed with 0.125 mL DPPH and 0.375 mL methanol and incubated for 0.5 h. The optical density was measured at λ_max_ = 517 nm. Ascorbic acid was used as a positive control. Radical scavenging activity was calculated as the following formula:% radical cation scavenging activity = (Abs control − Abs extract/Abs control) × 100(1)

#### 2.6.2. ABTS Radical Cation Scavenging Activity Assay

The antioxidative activity of the drug-free SNEDDSs was also determined using the ABTS radical cation scavenging assay as followed by Fahad et al., 2020 with minor modification [29,30]. Aqueous solutions of ABTS (7 mmol/L) and potassium persulfate (2.45 mmol/L) were freshly prepared. The two solutions were mixed after 12 h standing in the dark and incubated for 0.5 h, followed by standing in the refrigerator for 24 h. To the ABTS solution, a 50 µg/mL (1:1) differently prepared concentration of the formulation solution (10, 50, 100, 500, and 1000 μg/mL) was added to initiate the reaction for constructing a calibration curve. The absorbance was read at λ_max_ = 734 nm using a UV-Vis spectrophotometer. The ABTS (50 µg/mL) solution was used as a control. Ascorbic acid was used as standard. Three replicates for standard and extract were used for analysis. The percentage of antioxidant capacity for SNEDDSs was estimated based on the reduction in ABTS absorbance by using the following formula:% radical cation scavenging activity = (Abs control − Abs extract / Abs control) × 100(2)

### 2.7. Solid-State Characterization

#### 2.7.1. Modulated Differential Scanning Calorimetry (mDSC)

Pure crystalline APIs, SNEDDSs prepared according to the ‘Preparation of SNEDDS formulation section’, and physical mixtures (PMs) for the filament preparation were subjected to mDSC (DSC Q20, TA^®^ instruments, New Castle, DE, USA) analysis to identify the thermal events and their solid state. This preliminary data was used to decide the processing conditions for further experiments. Briefly, 5 to 15 mg of the samples were weighed and run from 35 °C to 200 °C, with a ramp rate of 5 °C/minute and modulated every 30 s for ±1 °C. The data collected were analyzed and presented as plots of reverse heat flow (mW) versus temperature (°C).

#### 2.7.2. Thermogravimetric Analysis (TGA)

Pure crystalline APIs and PM for filament preparation were subjected to a heating ramp at 10 °C/minute from 35 °C to 350 °C to obtain the thermal properties of the samples. A Mettler-Toledo TGA/DSC 1 (Mettler-Toledo, Schwerzenbach, Switzerland) was used for this purpose. Ultra-purified nitrogen gas was used as the purge gas at a flow rate of 50 mL/min. The data were collected and analyzed using the STAR software. It was plotted as % weight loss versus temperature.

### 2.8. Hot-Melt Extrusion Process

The preliminary screening of polymers for developing a carrier system for the oral delivery of the SNEDDS was conducted using thermal analysis of the excipients (differential scanning calorimetry). The thermal investigation was used to select the processing parameters, as mentioned above. These are shown in Figure 1. A physical mixture of HPC and Soluplus in a ratio of 1:1 was prepared and mixed before feeding to the extruder. A calibrated volumetric feeder (Brabender twin-screw feeder with stirring agitators, Brabender Technologie, Duisburg, Germany) was used to introduce this polymer blend to the extruder. A Leistritz ZSE 12 HP-PH 12 mm twin-screw corotating extruder with eight individual zones was used for the HME process. An optimized HME process was used to extrude filaments for developing the carrier tables for encapsulating the SNEDDS. The die used for the extrusion process had a diameter of 1.85 mm. The filaments were found to have a slightly larger diameter (1.90 ± 0.5 mm) due to expansion after extrusion. The feed rate was set at 2 g/min and the maximum torque was observed to be 36.0 % at equilibrium. The rpm for the extrusion was set at 50 rpm. The collection of filaments was started after the system reached a steady state and the materials extruded prior to reaching the steady state were discarded. The filament was evaluated for diameter uniformity post-manufacturing using a vernier caliper with measurements conducted at specified time intervals to ensure consistency in the diameter and later stored in a vacuum desiccator at 25 °C. The collection of the filaments was stopped after the feed stopped to avoid collecting post-equilibration step and maintained the same quality throughout. The filaments obtained had optimum properties for FDM 3D printing, as observed during the printing process.

### 2.9. Fused Deposition Modeling-Based 3D Printing

Solidworks (Dassault Systèmes, Vélizy-Villacoublay, France) was used to design the hollow tablet containers, and the model was designed in a cylindrical shape with an outer diameter of 12 mm and an inner diameter of 11.6 mm and 11 mm. The wall diameter was kept at 0.4 mm and 1 mm and the height of the container was set at 10 mm. The dimensions were set to accommodate 500 mg of the formulated SNEDDS based on prior evaluations. The design was sliced with a layer thickness of 0.1 mm, 100% infill density, and grid in-fill pattern, and saved as a .stl file with a tolerance of 0.0090 mm and 10 deg for the deviation and angle, respectively, for the hollow designs with 452 and 460 triangles for a 1 mm wall-thickness and 0.4 mm wall-thickness design. The objects were printed using an FDM 3D printer (Ultimaker S3, Utrecht, The Netherlands) at a temperature of 150 °C at a print speed of 50 mm/s. The bed temperature was kept at 80 °C and a 0.4 mm BB4 print head was used for the process [22,23,24]. The printing temperature was selected based on the preliminary thermal analysis, to ensure that the contents of our product to be filled in are not degraded by the heat of the nozzle. This was considered while ensuring that we could smoothly print the structure without any nozzle clogging or zone expansion. Prior testing was conducted to observe the optimum printing temperature to avoid nozzle clogging starting from the maximum extrusion temperature. The process was paused using in-process controls to fill in the pre-weighted SNEDDS formulation using a pipette. The process was then resumed, and the top layer was printed to seal off the container and manufacture SNEDD-containing tablets for oral delivery [23].

### 2.10. Morphological Evaluation

#### 2.10.1. Optical Microscopy and Size Measurements

The diameter and height of the printed formulation were determined using a Neiko 01407A digital caliper (VWR1, Philadelphia, PA, USA). It was also used to determine the filament diameters. The final weights of the tablet were measured using a Mettler-Toledo ME-TE (Mettler-Toledo, LLC., Columbus, OH, USA) analytical balance. The samples were imaged using a Dino-Lite AM7391MZTL optical microscope (AnMo Electronics Corporation, New Taipei City, Taiwan).

#### 2.10.2. Scanning Electron Microscopy (SEM)

The empty hollow prints were used for imaging using an FEI Quanta 650 ESEM to observe the surface morphology of the empty prints. Briefly, the samples were mounted on aluminum stubs with double-sided carbon tape and sputter-coated for 1 min using Pt/Pd in a Cressington sputter coater 208 HR (Cressington Scientific Instruments Ltd., Watford, UK)), and then observed using the Quanta SEM. The images were captured at a 52× to 53× magnification at an acceleration voltage of 15.00 kV.

#### 2.10.3. Micro-CT Imaging

The CUR- and LNS-loaded SNEDDS-filled print with a 1 mm wall diameter was exposed to micro-CT analysis with the following scan parameters: Zeiss Flat panel, 60 kV, 6.5 W, 0.05 s acquisition time, 5 samples per view, detector 220.167 mm, source −26.173 mm, XYZ [3426, −23,590, 1767], angle ±180, 4001 views, no filter, dithering, and multi-reference.

### 2.11. In Vitro Dissolution Testing

A small-volume pH-shift dissolution study protocol was developed for evaluating the performance of these 3DP tablets containing SNEDDSs. USP type-II dissolution apparatus (Paddle type) was used for the same purpose. A small dissolution study was used to improve the detection of release from the sample due to the small amounts used for the study. The tablets were first exposed to 120 mL of hydrochloric acid (HCl)–potassium chloride (KCl) buffer (pH 2.0, 0.1 M) for 2 h. This was followed by the addition of 30 mL of phosphate buffer (pH 6.8, 0.1 M) to produce a total volume of 150 mL, causing a pH shift. The dissolution system (Vankel VK7000, Agilent Technologies, Santa Clara, CA, USA) was maintained at 37 °C and stirred at 50 rpm. The media was manually collected at time intervals of 5, 10, 15, 30, 60, 120, 125, 130, 135, 150, 180, 240, and 360 min and replaced with fresh media. A total of 1 mL of this was filtered using a 0.45 µm filter. The filtered sample was diluted two folds using ACN and the concentration of the drug present in the sample was analyzed using an in-house-developed HPLC method (described below). The same protocol was followed for all the SNEDD-containing tablets and the study was conducted using 3 replicates (*n* = 3).

The concentration of drugs in the samples collected from the dissolution studies was analyzed using a reverse-phase high-performance liquid chromatography (RP-HPLC) analysis (Thermo Fisher Vanquish HPLC system, Thermo Fisher, Waltham, MA, USA) using a 25 cm × 4.6 mm, 5 µm particle size, stainless steel C-18 column (Discovery C-18 column, Supelco series, Millipore Sigma, Burlington, MA, USA). The mobile phase was prepared using deionized water with 0.1% ortho-phosphoric acid (pH 2.5) as the aqueous phase and acetonitrile (ACN) as the organic phase at a ratio of 90:10. The flow rate was set to 1 mL/min and the injection volume was set at 20 µL. The run time for each sample was 8 min. The single method used for detection showed a retention time of 3.3 min for curcumin and 2.8 min for lansoprazole. The detector used in this analysis was an ultraviolet-visible (UV-Vis) spectrophotometer (Thermo Fisher Vanquish HPLC system, Thermo Fisher, Waltham, MA, USA). The wavelength of detection was set at 260 nm. The estimation of the collected data was performed using a calibration curve ranging from 1 to 128 µg/mL (R^2^ = 0.999).

## 3. Results

### 3.1. Formulation Development of SNEDDSs and Drug Loading

The representative SNEDDS drug solubility was found to be as high as 13.6 and 40.5 mg/g for LNS and CUR, respectively. However, to avoid the risk of precipitation, 10 mg (74% of the equilibrium solubility) of LNS and 20 mg (49% of the equilibrium solubility) of CUR were loaded on the dosage forms and utilized for further formulation-preparation steps.

### 3.2. Formulation Droplet Size and PDI Analysis

The droplet-size analysis of the drug-free and drug-loaded SNEDDS formulations showed slightly different particle sizes upon aqueous dilution. The drug-free formulation produced a droplet size of 75.7nm, whereas the drug-loaded SNEDDS yielded a slightly smaller droplet size of 70.9 nm, respectively (Table 2). The polydispersity index (PDI) of the SNEDDSs were within the range of 0.52–0.53. Both drug-free and drug-loaded SNEDDSs showed accepted zeta potential values (−15, −19 mv) (Table 2).

### 3.3. Assessment of Self-Emulsification Efficiency of Bio-SNEDDSs

The assessment test showed that the representative SNEDDS formulation was homogeneous, transparent, and spontaneously dispersed upon aqueous dilution. Figure 2 represents the appearance of the SNEDDS formulations after aqueous dilution (1 in 1000 *w*/*w* ratio) with water at room temperature (25 °C) after the initial and 6 months.

### 3.4. Transmission Electron Microscopy (TEM)

Transmission electron microscopy (TEM) is a very important technique used for investigating the shape of the microstructures with high-resolution images and it can capture any transitions of the structure. The images from the TEM analysis (Figure 3) show that the droplets of the drug-free SNEDDSs have a spherical structure and similar sizes, which was also confirmed by the particle size analysis. The droplet sizes were found to be uniform around 65–100 nm in the nano-emulsifying systems, which strongly correlates to the droplet-size study findings.

### 3.5. Antioxidant Studies

The representative SNEDDSs showed dose-dependent scavenging activities in both DPPH radical and ABTS radical cation scavenging activity studies (Table 3). At maximum concentration (1000 µg/mL), SNEDDSs showed close activities to the positive control (ascorbic acid). The concentration required to result in a 50% antioxidant activity (IC_50_) of the represented SNEDDSs was 330.1 µg/mL in the DPPH method, while it was 393.9 µg/mL in the ABTS method (Figure 4).

### 3.6. Solid-State Analysis

Solid-state thermal analysis of the pure drugs showed that curcumin and lansoprazole were stable up to temperatures of 220 °C and 175 °C, respectively (Figure 5). The mDSC analysis showed characteristic melting points at 173 °C and 175 °C for lansoprazole and curcumin, respectively, identified by the thermal transitions (Figure 6). However, the melting point and degradation for lansoprazole were very close to each other, i.e., it melted with degradation. The characteristic thermal transitions of CUR and LNS were absent in the tested SNEDD samples, indicating a change from the crystalline state of the drug to its amorphous state during SNEDD formation. Additionally, a small, broad thermal transition can be observed in the case of the curcumin and lansoprazole pure-drug physical mixture at a temperature of 152 °C. This transition observed much earlier than the individual melting point of the two drugs might be due to some interaction between the two drugs, and needs to be investigated separately.

### 3.7. Preparation of Filaments for Preparing SNEDDS-Carrier Tablets via HME

A 50–50 (% *w*/*w*) mixture of HPC and Soluplus was used to prepare the 3D-printable filaments that were used to prepare the container system for achieving the desired release profiles. A 50–50 mixture of HPC and Soluplus produced good 3D-printable filaments with optimum physical and chemical properties using the HME process. Even though a 1.85 mm die was used for the extrusion process, the collected filaments were found to be around 2.01 ± 0.7 mm. This may be attributed to polymer swelling as soon as it comes out of the extruder. The filament sample was collected when the extrusion reached the steady state and the die pressure and torque had reached a steady level as well.

### 3.8. 3D Design and FDM Printing

The design was constructed using the 3D-builder software, as previously mentioned. The tablet in Figure 7 and Figure 8A shows a hollow top, which was then sealed and can be observed in the adjacent Figure 8B. It was sliced using the Cura software and printed with skirt support, which helped in its easy removal and aided to hold its shape. The designs printed were smooth and of a uniform shape and size with no degradation of the filament observed during the printing process. As the 3D printer has no mixing or kneading sections, a higher printing temperature was needed to print and smoothly extrude the molten filament through the nozzle. The printing temperature for the current study was 150 °C. Printing was also tried at higher temperatures, but the filaments showed discoloration at temperatures over 210 °C. Additionally, even at a higher temperature, no difference was observed in the printing quality, and hence the printing was conducted at the lowest temperature possible. 

### 3.9. Tablet Morphology

The FDM 3D-printed tablet containing a SNEDDS had a uniform geometry with a diameter of 12 mm and a height of 10 mm. The side view of the tablet shows a small band of sealing where the print was paused to fill in the SNEDDSs and later sealed off (Figure 8B, red arrow). The top view of the print captured using the digital camera shows a blip indicating the tear-off of the filament at the end of the printing process (Figure 8A, red circle). A slight expansion of the diameter was observed at the bottom of the tablet, which may be attributed to the load put on the lower layer by the consecutive layers printed later. This phenomenon was not observed as we moved up the tablet layers and a uniform diameter was observed. The quality of the print was assessed to ensure that there were no leaks from the system. Initial tests were conducted by using a dye solution filled in the system and immersion in water for a minute to observe leaks (data not shown).

SEM images show a hollow print morphology with the wall rise starting towards the edge and an empty section at the base (Figure 9A), as designed in the 3D-design model. The surface morphology of the top surface (Figure 9B) shows the well-connected print lines ensuring no leaks from the print post-sealing step.

Micro-CT imaging showed a step-by-step buildup of the SNEDDS-filled print, which produced a clear indication of the hollow print being filled and sealed. The initial crosslines in Figure 10B show the support structure printed to support the hollow print from collapsing on the inside during the printing process. This base layer, when completed to 80% post-filling, can be observed in Figure 10C, with the hollow sections being occupied by the filled SNEDDSs. The complete sealing of the tablet can be observed in Figure 10D, which aligns with the images captured using SEM and optical microscopy.

### 3.10. In Vitro Drug-Release Testing

The formulated containers encapsulating CUR- and LNS-containing SNEDDSs were subjected to dissolution studies to understand the effect of encapsulation in a polymeric container on the drug release of the SNEDDS formulation inside. The wall diameter was varied to 0.4 mm and 1 mm thicknesses to understand the effect of the tablet disintegration and release mechanism on the same. Both drugs released from the system were evaluated using the developed HPLC method for analysis. As observed in Figure 11 and Figure 12, during the release studies, pure SNEDDSs showed an immediate drug release in the case of both drugs, as compared to the release from the container system. The 3D-printed matrix was developed using an HPC matrix, which tends to release drugs via a swelling mechanism. This release is associated with a certain lag time for the matrix to swell before the start of water ingress followed by release. As observed in the case of both the drugs, the 0.4 mm wall being thinner than the 1 mm counterpart design had a shorter lag time and started releasing the SNEDDSs contained in the system at around 30 min. However, the release was found to start only after 90 min in the case of the designs with a thicker wall. All the systems were able to achieve a drug release of almost 85% in the case of curcumin, and approximately 18% in the case of lansoprazole. The lower drug concentration, in this case, can be attributed to the poor inherent solubility of the drug. No surfactants were used in the dissolution media and the study was conducted in non-sink conditions to study the effect of wall thickness and SNEDDS composition on release.

Post-dissolution imaging (Figure 13) of the tablets shows a yellow band on these tablets, indicating the release of curcumin (as identified by the yellow color). These tablets were found to be swollen and sticky due to the hydrophilic composition of the feedstock filaments.

## 4. Discussion

Proper knowledge and selection of the right excipients are important for the successful formulation design in a lipid-based system [31]. Solubility is an important parameter for any drug formulation because it offers the required information for the maximum dose that can be included in a single-unit dosage form. Particle size is another important parameter that can play a key role in the oral absorption of the drug in vivo. The smaller the droplet size, the larger the interfacial surface area that is provided for drug absorption, although it should be recognized that the dispersion may be substantially modified in real-time by the biological products induced for digestion [32]. The efficient SNEDDSs spontaneously form a nano-sized oil-in-water (o/w) emulsion upon aqueous dilution with mild agitation (a very low-energy input). In the optimization process, if the formulation is clear, homogeneous, and took less time (i.e., less than 1 min), it is efficient within the scope of the current studies.

In this study, a bioactive oil, namely, black seed oil (BSO) was used with surfactant/cosolvent to develop the optimized formulation. The formulation was optimized in terms of its appearance upon aqueous dispersion, droplet size, and drug-loading capacity. The SNEDDSs containing BSO and Capmul MCM (mono and diglycerides) with Kolliphor ELP (HLB 14–16) showed balanced characteristics between high drug solubility, small droplet size, and good self-emulsification properties. Accordingly, this formulation was utilized in subsequent 3D-printing studies.

The solid-state thermal analysis reinforced the feasibility of the 3D-printing conditions with no expected degradation of CUR or LNS at our working temperature range, which would have been made possible by the heat of the FDM printer nozzle (150 °C) coming into contact during the sealing step. Of the standard nozzle sizes available, a 0.4 mm nozzle was selected, which helped in avoiding nozzle clogging and helped to print the HPC-Soluplus filaments without any issue. A container system containing SNEDDSs without any leaks was successfully prepared using this method. The in vitro drug-release studies showed enhanced CUR and LNS release upon decreasing the tablet’s wall diameter to 0.4 mm. This may be attributed to the faster swelling and subsequent release through the thinner hydrophilic polymer matrix that was used to produce the hollow tablet container. Thinner walls swell and erode faster, when compared to thicker walls. Additionally, the Soluplus present in the same aided in the degradation of these tablet containers. The print dimensions post-dissolution study (as observed in Figure 13) shows a reduced diameter and height, which can be attributed to the erosion of the print following the initial dissolution step. The print shows a smooth, swollen structure as opposed to the pre-dissolution testing shape observed in Figure 8. This characteristic change in shape depends on the interaction of the polymer chain interactions with the media, which, in turn, is dictated by the composition and polymer properties. The filament used for the preparation of the container system can be further explored to produce a more versatile system for delivering any additional drug [33]. This aids multi-drug regimens while incorporating different drug classes as well. A blank filament can be made with a composition to release the drug based at the site of drug absorption [34]. The use of pH-dependent release polymers can help deliver drugs that are susceptible to degradation in the acidic pH. The incorporation of a drug in the filament can be conducted, based on the intended therapeutic requirements [21].

Previous studies showed that a capsule-filled liquid SNEDDSs was able to provide complete (>80%) drug release within the first 30 min [9,12,35]. In particular, capsule-filled CUR-SNEDDSs showed >80% CUR release in the first 10 min of a relevant dissolution study [36]. In the current study, the 3D-printed tablets showed maximum releases of 85% and 18% for CUR and LNS, respectively, by the end of the dissolution study (at 6 h). These findings are in agreement with the previous studies that showed a significant delay in drug release from the 3D-printed SNEDDS tablets, compared to the conventional capsules filled with pure drugs [27]. Previous studies revealed a strong wetting behavior of the liquid SNEDDS formulation and the tablet shell resulting in a strong adhesive force between them, which decreased the drug-release rates from the 3D-printed tablet [27]. The wettability also affected the forces (capillary and diffusion), which caused the dissolution medium to penetrate the dosage form and the total drug-loaded liquid formulation to be released from the dosage form. During the study, the erosion of the tablet shell was proposed to contribute to the diffusion [37]. The pore structure in the shell is changed by erosion, which increases the porosity with increasing dissolution time. This further affects the interconnectivity of the pores, which eventually facilitate the formation of micro channels. These channels further increase in size, and a capillary force builds up in these channels causing a faster release process once the matrix starts swelling and eroding, i.e., past the lag time. The design of the shell or container system, in this case, is important. The slow erosion of the matrix may cause supersaturation inside the system due to high drug concentrations in low amounts of media. This may lead to crashing out and precipitation of the drug, leading to dose dumping or ineffective dosing.

At the beginning of the dissolution study, the 3D-printed tablets were found to float on the top of the dissolution media and slowly sink until the end of the study. This phenomenon is attributed to the low density of SNEDDSs present within these tablets. Similar findings were observed in previous studies on 3D-printed SNEDDS tablets [24]. An erratic release may be due to the floating of the tablet that led to unequal exposure to dissolution media. Another reason could be due to the supersaturated state being formed inside the container system from water penetration through the swollen polymeric walls. SNEDDSs need a large quantity of water to disperse and form uniform and dispersed nanoparticles in situ. However, the slow influx of water leads to the saturation of the SNEDDS system inside, thus altering the release from the formulated SNEDDSs. This drug, then free from the particulate SNEDDSs, must further diffuse through the eroded walls of the polymeric carrier and reach the dissolution media. Thus, this incorporation led to an increase in the number of barriers when compared to the release of the drugs from SNEDDSs themselves.

Another critical parameter that should also be considered is the difference between 3D-printed and conventional compressed tablets in manufacturing techniques. During compaction, tablet constituents are subject to plastic and elastic deformation before the formation of inter-particulate bonds [38]. The reversible viscoelastic process of deformation, i.e., strain recovery, is one of the key mechanisms involved in the disintegration and erosion process of the compressed tablet upon exposure to GI fluids [39,40]. On the other hand, 3D-printed tablets lack particle deformation as no compression takes place. Accordingly, strain recovery is not expected to have a significant effect on drug-release behavior from 3D-printed tablets. The drug release from 3D-printed SNEDDS tablets is influenced by several factors, such as the physical properties of drug SNEDDS and polymer SNEDDS, the microstructural properties and thickness of the HPC-Soluplus tablet shell, the properties of the dissolution medium (such as temperature and pH), and properties of the fluid/tablet wall (e.g., wettability/contact angle) [41]. Previous, studies have shown the incorporation of a solidified lipid matrix inside a 3D-printed system [42]. However, we expanded the scope of a multi-drug delivery regiment using a lipid-based formulation through our study. Semi-solid 3D printing of just lipid-based systems had poor printing fidelity and lacked consistency, which can be overcome by using the method employed in this study [24,26,42].

This study shows the capability to deliver contents irrespective of their physical state to the site of action within the GIT. Three-dimensional printing provides the advantage of personalizing the drug-delivery approach for the patient under treatment. The delivery of biologics via an oral route would be possible using this approach, while encasing it inside a suitable system that prevents its degradation and maintains the integrity and stability. The conversion of large molecules, proteins, and biologics to solid powders using lyophilization techniques lead to loss inactivity and, therefore, most biologics are administered via intravenous routes or formulated into lipid nanoparticles [43,44,45,46]. These liquid-phase systems can be delivered similarly, as demonstrated in this study.

## 5. Conclusions

The SNEDDSs showed balanced characteristics between high drug solubility, small droplet size, and good self-emulsification properties. The solid-state thermal analysis reinforced the feasibility of the 3D-printing conditions with no expected degradation of CUR or LNS at our working temperature range to which our printed constructs would have been exposed by the heat exerted from the FDM printer nozzle (150 °C) coming into contact during the sealing step. The filament used for the preparation of the container system can be further explored to develop a more versatile system for delivering an additional drug. Combining these two drug-delivery platforms further opens more options for tackling a multitude of issues related to multi-drug delivery as well as the delivery of drugs susceptible to degradation. A lot of factors are involved in the system design and selection of drug-excipient candidates. These influence the final behavior of the formulation. More research is warranted to thoroughly investigate the drug-release mechanisms occurring in 3D-printed dosage forms. A comprehensive analysis of the drug-release mechanisms is essential for the future product design of 3D-printed SNEDDS dosage forms. The floating property plus the slow release from 3D-printed SNEDDS tablets can be used to deliver the SNEDDS in a sustained release manner, which is difficult to achieve with direct SNEDDS delivery. We thus showed an innovative approach to deliver personalized medications for multi-drug containing difficult-to-deliver regimens, while combining different formulation approaches.

## Figures and Tables

**Figure 1 pharmaceutics-14-01082-f001:**
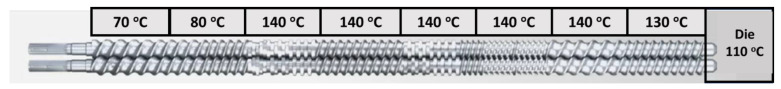
Screw design used for processing of filaments using a hot-melt extrusion process.

**Figure 2 pharmaceutics-14-01082-f002:**
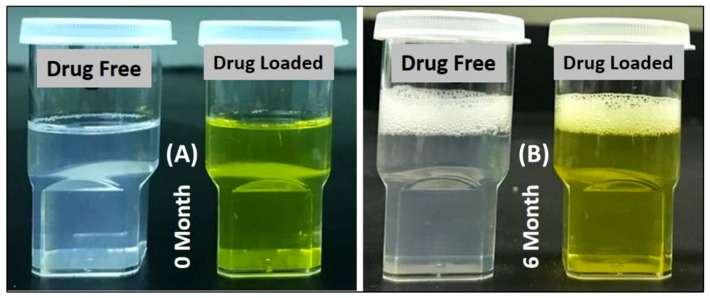
Morphology of the drug-free and drug-loaded SNEDDSs when 100 mg of blank formulation was diluted in 100 mL of water in (**A**) 0 months and (**B**) after 6 months of storage.

**Figure 3 pharmaceutics-14-01082-f003:**
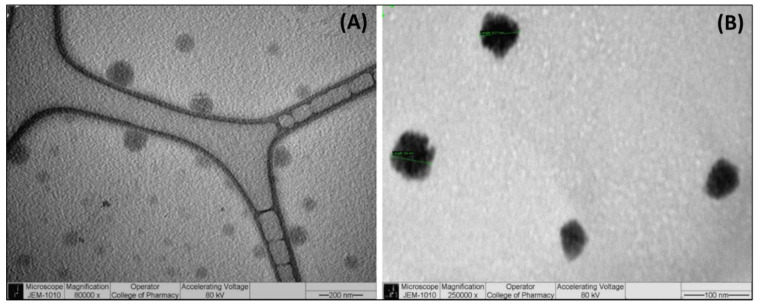
TEM images showing the structure and shape of the droplet sizes of the SNEDDSs at (**A**) 80,000× and (**B**) 250,000× magnifications.

**Figure 4 pharmaceutics-14-01082-f004:**
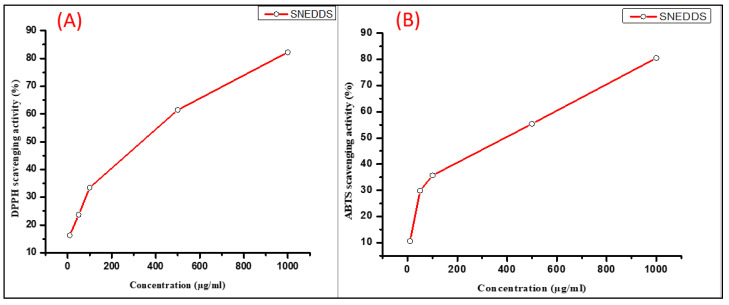
Antioxidant activities of the representative SNEDDSs. (**A**) DPPH scavenging activity and (**B**) ABTS scavenging activity. Data were analyzed using Origin Pro 8 software.

**Figure 5 pharmaceutics-14-01082-f005:**
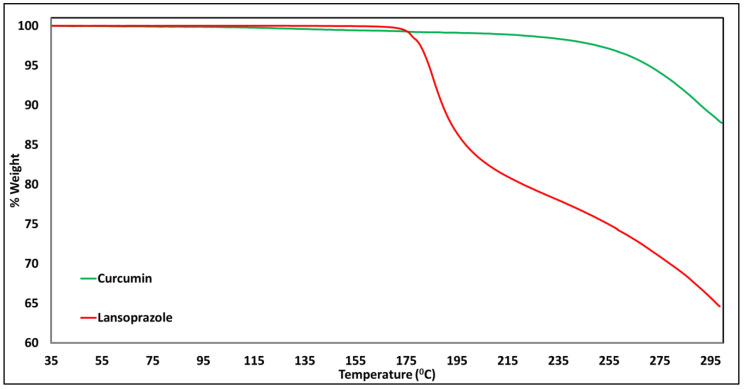
Thermogravimetric analysis of curcumin and lansoprazole pure drugs.

**Figure 6 pharmaceutics-14-01082-f006:**
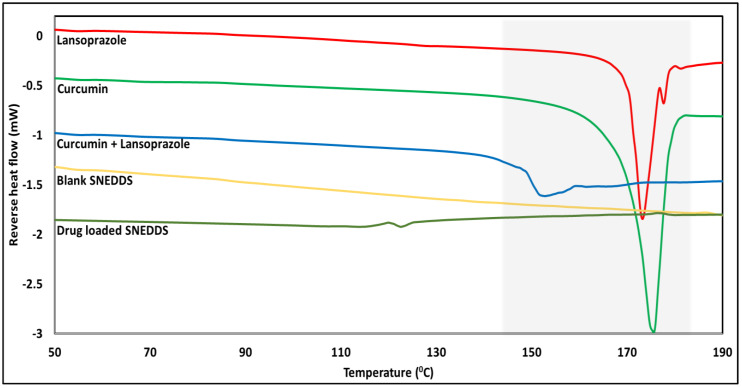
Modulated differential scanning calorimetry profiles for SNEDDS evaluation.

**Figure 7 pharmaceutics-14-01082-f007:**
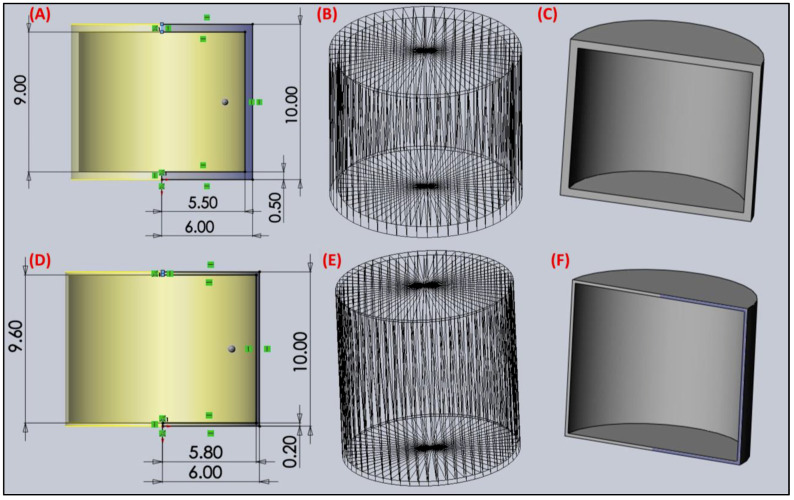
3D designs developed for printing the hollow container system: (**A**) dimensional measurements, (**B**) .stl file triangle count and (**C**) cross-sectional view for the print with 1 mm wall thickness, (**D**) dimensional measurements, (**E**) .stl file triangle count, and (**F**) cross-sectional view for the print with 0.4 mm wall thickness.

**Figure 8 pharmaceutics-14-01082-f008:**
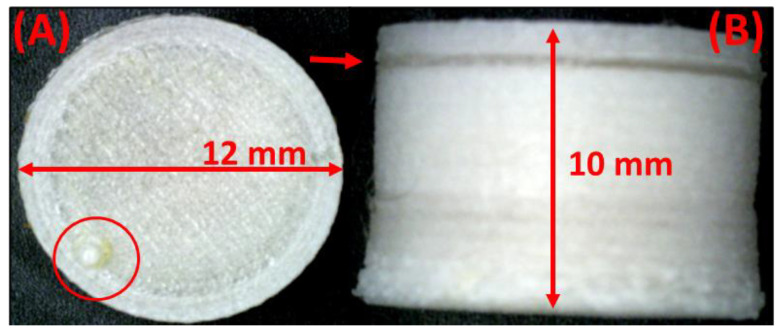
Tablet images captured using an optical microscope: (**A**) top-view of a tablet with 1 mm wall diameter and (**B**) side-view of the tablet with 1 mm wall diameter.

**Figure 9 pharmaceutics-14-01082-f009:**
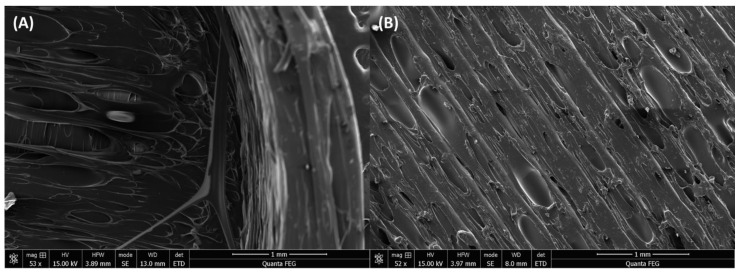
Scanning electron microscopy (SEM) images of hollow print showing: (**A**) the edge of the internal structure, and (**B**) the surface morphology of the sealed top.

**Figure 10 pharmaceutics-14-01082-f010:**
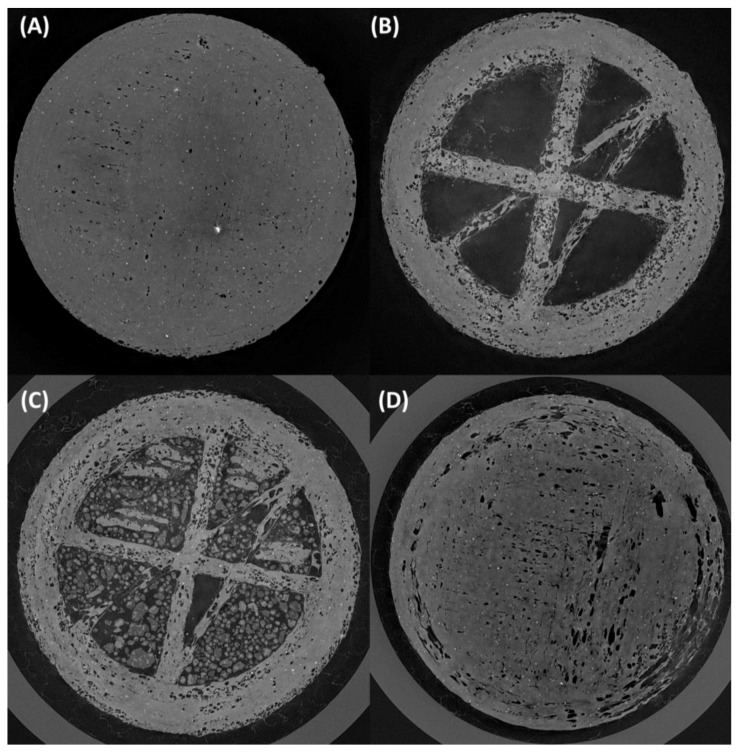
Micro-CT images of CUR- and LNS-loaded SNEDDS-filled prints: (**A**) first printed layer, (**B**) cross support printed after the first layer to prevent collapsing of the hollow print during the additive manufacturing process, (**C**) SNEDD-loaded prints at 80% with hollow space occupied by the drug-loaded SNEDDS, and (**D**) top layer post the sealing process.

**Figure 11 pharmaceutics-14-01082-f011:**
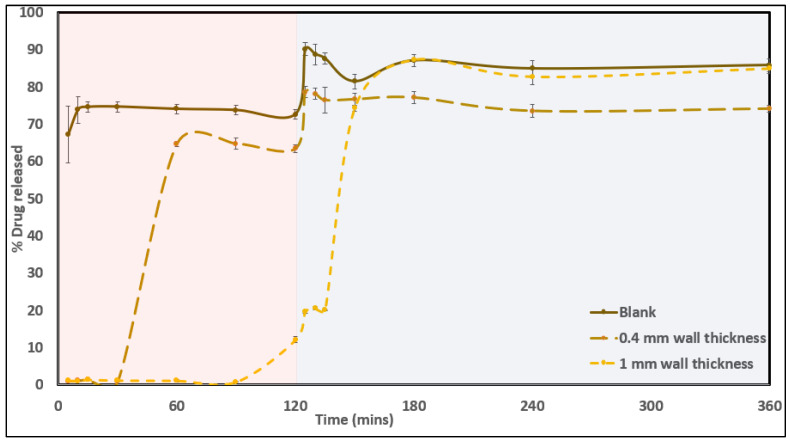
In vitro drug-release profile for curcumin from pure SNEDDS, SNEDDS tablets from 1 mm and 0.4 mm wall-diameter designs.

**Figure 12 pharmaceutics-14-01082-f012:**
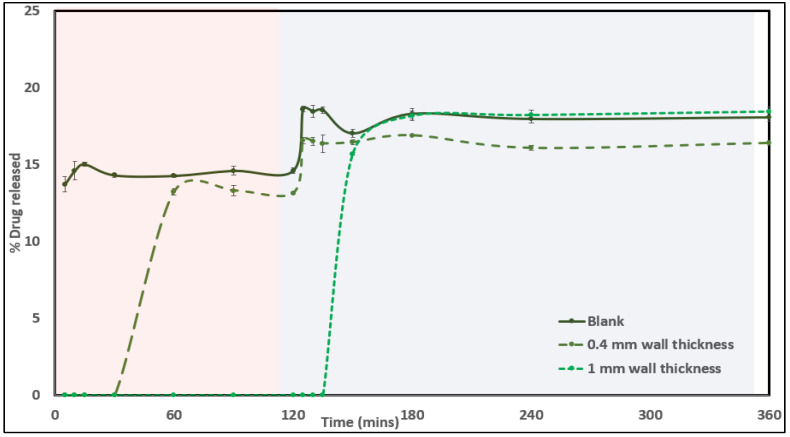
In vitro drug-release profile for lansoprazole from pure SNEDDS, SNEDDS tablets from 1 mm and 0.4 mm wall-diameter designs.

**Figure 13 pharmaceutics-14-01082-f013:**
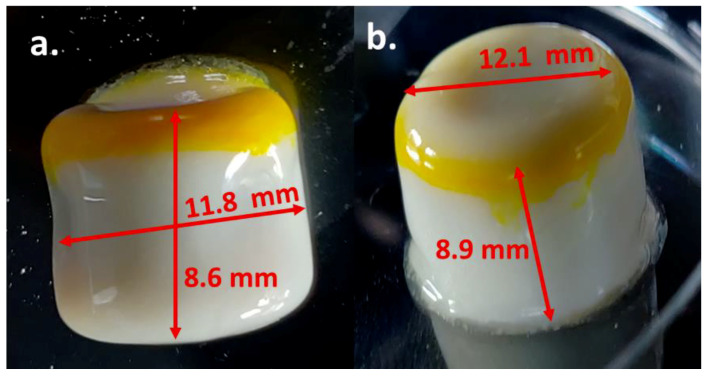
Post-dissolution images of prints containing SNEDDS (**a**) print with 0.4 mm wall thickness and (**b**) print with 1 mm wall thickness.

**Table 1 pharmaceutics-14-01082-t001:** Composition of SNEDDSs and their drug-loading capacity (saturated solubility).

Formulation	BSO	CMCM	TCP	KELP	Total
**SNEDDS**	25%	20%	5%	50%	100%
**Composition (% *w/w*)**				**Saturated solubility (mg/g)**	
				**LNS**	**CUR**
BSO/CMCM/TCP/KELP (25/20/5/50)				13.59 ± 0.39	40.49 ± 1.33
				**Drug loading in dosage form (mg/g)**	
BSO/CMCM/TCP/KELP (25/20/5/50)				10.00	20.00

**Table 2 pharmaceutics-14-01082-t002:** Droplet size, polydispersity index, and zeta potential values of the drug-loaded and drug-free SNEDDSs.

Formulation	Particle Size (nm)	Zeta Potential (mV)	PDI
Drug-loaded SNEDDS	70.90 ± 0.17	−19.1 ± 2.26	0.534 ± 0.023
Drug-free SNEDDS	75.74 ± 1.81	−15.6 ± 4.39	0.519 ± 0.016

**Table 3 pharmaceutics-14-01082-t003:** Scavenging activities of the representative SNEDDSs.

Sample	(DPPH Radical Scavenging Activity in %)				
(µg/mL)	10	50	100	500	1000
** *SNEDDS* **	16.3 ± 4.4	23.7 ± 1.7	33.4 ± 1.1	61.4 ± 3.2	82.2 ± 3.7
**Ascorbic acid**	80.7 ± 2.0	85.1 ± 1.3	85 ± 1.2	88.7 ± 2.4	90.7 ± 1.4
	**(ABTS Radical Cation Scavenging Activity in %)**				
** *SNEDDS* **	10.6 ± 0.3	29.9 ± 1.6	35.7 ± 2.0	55.4 ± 2.1	80.5 ± 3.1
**Ascorbic acid**	80.7 ± 2.4	81.2 ± 2.1	84.2 ± 1.9	87.2 ± 2.4	88.7 ± 2.1

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
