# Peer review of "Three-Dimensional Printing of a Container Tablet: A New Paradigm for Multi-Drug-Containing Bioactive Self-Nanoemulsifying Drug-Delivery Systems (Bio-SNEDDSs)"

_pharmaceutics, 2022, doi:10.3390/pharmaceutics14051082_

Round 1

Reviewer 1 Report

The manuscript describes the integration of SNEDDS into a 3D printed tablet to control the rate of loaded drug release. The manuscript starts by describing the synthesis of SNEDSS and then explains how they were integrated into the 3D printed compartment along with the stability and dissolution studies. The article is well presented and suitable for publication in pharmaceuticals after considering few minor revisions

  • Authors need to explain the difference between SEDDS, SMEDDS and SNEDDS as they discuss these in the introduction.
  • Figure captions need to be more detailed for example figure 3 captions does not explain the difference between the figure on the left-hand side and right-hand side. Ideally these has to be labelled separately as 3A and 3B. Similarly, fig 2 captions and labelling need to have ore details.

Reviewer 2 Report

Dear Authors,

The article presents interesting research results but in my opinion it could be improve. I recommend few changes:

1. I recommend in introduction add information about printing parameters and it influence on 3d printing fdm process (e.g.

ANALYSIS OF THE DIMENSIONAL ACCURACY OF CASTING MODELS MANUFACTURED BY FUSED DEPOSITION MODELING TECHNOLOGY,
Surface Texture of Models Manufactured by FDM Technology, 
DOI 10.1063/1.5056274

2. Moreover I recommend to in point 2.8 explain in more depth way the results of producting filament e.g. it quality dimensional deviation etc.

3. In point 2.9 you could show the virtual building platform with samples on it.

4. In the same point 2.9 please add more information about stl file, what was the tolerance and save parameters how stl file looks like, maybe add photos or describe number of triangles and deviation set.

5. In point 2.9 please explain why did you choose temperature 150 degrees, for new materials it reguire a lot of tests to determine temperature, so please explain why did you choose these one.

6. In points 3 you desctibe results but could you please explain what was the assess of the quality of printed samples? any measurements after printing if yes maybe it would be helpfull to presents it in tables.

7. In point 3.8 you show the samles on the platform maybe it would be good to do this for the rest of samples.

8. In point 4 Discussion I recommend to focus on the shape and size of printed samples and samples after all other postprocessess showed on figure 11.

9. Conclusion should be longer and supported by the results at least related to the dimension and how postprocessing change it.

The article can be published after all of these improvements.

Regards,

Reviewer

Reviewer 3 Report

The authors propose a new strategy to create efficient multi-drug delivery systems as well as delivery of drugs susceptible to degradation that combines 3D printing (fused deposition modeling in this case) with self-nano emulsifying drug delivery systems (SNEDDS).

The technical approach developed here is highly interesting and promising. The authors offer a complete study from the optimization of the SNEDDS formulation and the adjustment of the 3D printing parameters to in vitro dissolution testing. They conclude that the system is capable of delivering contents that are independent of their physical state.

The missing element in this work is the study the 3D printing model. Only images under optical microscopy are provided. Micro Computed Tomography could be used to examine the porosity and interconnectivity of the tablet. Scanning Electron Microscopy could be used to discuss the morphology as well as the slight expansion the authors mentioned. I consider this work suitable for publication only after appropriately addressing this point.
